# The Evolution of Complex Muscle Cell In Vitro Models to Study Pathomechanisms and Drug Development of Neuromuscular Disease

**DOI:** 10.3390/cells11071233

**Published:** 2022-04-05

**Authors:** Jana Zschüntzsch, Stefanie Meyer, Mina Shahriyari, Karsten Kummer, Matthias Schmidt, Susann Kummer, Malte Tiburcy

**Affiliations:** 1Department of Neurology, University Medical Center Goettingen, 37075 Goettingen, Germany; stefanie.meyer@med.uni-goettingen.de (S.M.); karsten.kummer@med.uni-goettingen.de (K.K.); matthias.schmidt6@stud.uni-goettingen.de (M.S.); 2Institute of Pharmacology and Toxicology, University Medical Center Goettingen, 37075 Goettingen, Germany; mina.shahriyari@med.uni-goettingen.de; 3DZHK (German Center for Cardiovascular Research), Partner Site Göttingen, 37075 Goettingen, Germany; 4Risk Group 4 Pathogens–Stability and Persistence, Biosafety Level-4 Laboratory, Center for Biological Threats and Special Pathogens, Robert Koch Institute, 13353 Berlin, Germany; kummers@rki.de

**Keywords:** myositis, organoid, tissue engineering, drug screening, vascularization, co-culture

## Abstract

Many neuromuscular disease entities possess a significant disease burden and therapeutic options remain limited. Innovative human preclinical models may help to uncover relevant disease mechanisms and enhance the translation of therapeutic findings to strengthen neuromuscular disease precision medicine. By concentrating on idiopathic inflammatory muscle disorders, we summarize the recent evolution of the novel in vitro models to study disease mechanisms and therapeutic strategies. A particular focus is laid on the integration and simulation of multicellular interactions of muscle tissue in disease phenotypes in vitro. Finally, the requirements of a neuromuscular disease drug development workflow are discussed with a particular emphasis on cell sources, co-culture systems (including organoids), functionality, and throughput.

## 1. Introduction

Neuromuscular diseases (NMDs) are a large group of rare diseases with an increasing number of distinguishable entities due to the efforts of next generation sequencing and the discovery of liquid and imaging biomarkers. NMDs are sub-classified in motoneuron diseases, (poly-) neuropathies, diseases of the neuromuscular junction, and myopathies. Myopathies encompass more than 1000 etiologies, resulting in either from acquired or genetic causes. The unifying clinical symptoms are muscle wasting and weakness often accompanied by myalgia. The quality of life is highly impaired in patients with myopathies, mostly related to impaired mobility.

In this review, we focus on idiopathic inflammatory myopathies as one exemplary disease for myopathies to summarize how recent developments in muscle models may be utilized to study disease mechanisms and to identify and validate therapeutic targets in vitro.

Idiopathic inflammatory myopathies (IIM; synonym: non-infectious myositis, short myositis) are assigned as acquired myopathies and are a heterogeneous group of autoimmune diseases affecting multiple organ systems defined by skeletal muscle involvement and categorized into five major groups: dermatomyositis (DM), polymyositis (PM), immune-mediated necrotizing myopathy (IMNM), overlap syndrome with myositis (OM) including antisynthetase syndrome (ASS), and inclusion body myositis (IBM) [1]. The subclassification is based on a combination of clinical, serological, imaging, and histological features [2] (for details see Review [3]).

In order to better understand the cellular components, cell types, and necessary supportive molecules that may be relevant to modeling disease-specific organoids, we will briefly review the typical histopathological changes of IIM.

The muscle histology of all IIM shares inflammatory changes, but each subtype differs in the extent and localization of major histocompatibility complex (MHC) class I or II upregulation, infiltration of various immune cells (e.g., T, B cells, dendritic cells, and macrophages) and staining patterns for specific pro-inflammatory markers (e.g., molecules of the interferon signature). As an example, the core biopsy feature of DM consists of a perifascicular pathology with signs of perifascicular atrophy, CD56, and neonatal Myosin heavy chain (nMyHc) upregulation, an increased MHC class I staining signal, and complement deposition on endomysial capillaries and small blood vessels [4,5]. Of note, perifascicular atrophy affects both type 1 and type 2 muscle fibers [6]. The inflammatory infiltrates in DM are predominantly localized at perivascular sites and to a lesser extent in the endomysium, and are composed largely of B cells accompanied by CD4^+^ T helper cells, dendritic cells, and macrophages [7]. In comparison, ASS also exhibits perifascicular pathology, but with characteristic perifascicular necrosis and strong MHC class I and MHC class II upregulation [5,8]. IBM muscle tissue shows degenerative features such as amyloid deposition, vacuoles, tubulofilaments, and mitochondrial damage in addition to the inflammatory changes [9]. In INMN muscle tissue, muscle fiber necrosis is the most prominent feature while inflammation is relatively rare [7]; when detectable, necrotic fibers can be invaded by macrophages [10].

Despite careful histological analyses, the pathogenic mechanisms by which IIM develop remain poorly understood and vary by subtype [11,12]. Several environmental and genetic risk factors are discussed as possible initiators of autoimmunity, with the latter providing evidence for common as well as unique innate and adaptive immune and non-immune pathways [13].

Muscle fibers in IIM play a crucial role in this autoimmune process by abundantly expressing MHC class I molecules and innate immune receptors that can be activated by danger-associated molecular patterns and cytokines [13,14,15]. In addition, myofibers are capable of activating NF-κB signaling, producing and secreting various pro-inflammatory cyto- and chemokines in order to attract the appropriate immune cells into a milieu that is already preconditioned for antigen processing and presentation by dendritic cells [16,17].

Here we exemplify the role of adaptive immunity in IBM muscle tissue. The most abundant immune cells are T cells, particularly granzyme B and performing-expressing CD8^+^ T cells which surround and invade MHC class I-expressing and non-necrotic muscle fibers where they expand clonally [18,19]. Some of these T cells, such as CD28null T cells, have the unique properties of being highly cytotoxic, apoptosis-resistant, and pro-inflammatory [20]. B cells and plasma cells are less present in the muscle of IBM patients, yet this indicates a humoral component to this disease supported by the presence of anti-cytosolic 5ʹ-nucleotidase 1A (cN1A) in 30–80% of IBM patients [21].

Given the focus on organoids, we do not go into detail about pro- and anti-inflammatory cytokines, CXC chemokines, and CC chemokine profiles expressed in the respective IIM phenotype (for details see review De Paepe 2015), but we will highlight some recently discovered specific signatures of the different groups of IIM through the deep phenotyping of muscle tissue that could be useful in advanced in vitro models to mimic disease subtypes.

Gene expression profiling of muscle biopsies from IIM patients in combination with a machine-learning algorithm showed that the novel gene markers CAMK1G (calcium/calmodulin-dependent protein kinase IG), EGR4 (early growth response protein 4), and CXCL8 (interleukin 8) were only present in ASS. Further identified genes were selectively expressed in IMNM patients with anti- HMGCR autoantibodies apolipoprotein a4 (aPoa4) or in anti-Mi2-positive DM (MaDcaM1) (mucosal vascular addressin cell adhesion molecule 1) [22]. Other research groups, also using transcriptomics, have demonstrated upregulated pathways related to cell adhesion molecules (Cadherin 1) [23] or disturbed calcium homeostasis, dysregulation of LCK, and associated deregulation of apoptotic control of T cells in IBM [24]. These findings may provide a valuable clue to the pathological mechanisms of IIM and underline the role of T cells in IBM.

Collectively, innate and adaptive immunity contribute to the ongoing pro-inflammatory environment in IIM and subsequent induction of a cell stress response, leading to defective autophagy [25], impaired proteasome function [26], mitochondrial abnormalities [27,28], and the elevation of the inducible nitric oxide synthase [29] which induces nitric oxide (NO)-stress. All these mechanisms may contribute to reduced muscle contractility.

For IIM, treatment with various anti-inflammatory substances is the gold standard and is usually effective in PM, DM, OM, and IMNM to different degrees. Although the inflammatory processes are rapidly diminished, the regain of strength is often delayed or even missing in refractory patients [30,31]. Moreover, immunosuppression with glucocorticosteroids (GS) or GS-sparing drugs causes unintended long-term side effects. In IBM, no efficient treatment is available yet, which could be explained by ongoing cell stress leading to defective protein homeostasis and a change in the metabolism of the muscle despite immunosuppression [32].

In addition to the classical immunosuppressive treatment, specifically targeting dysfunctional muscle regeneration is an additional therapeutic strategy in IIM to regain muscle function. Among several approaches, such as selective androgen regulators (testosterone) or ghrelin and its mimetics [33], drugs targeting myostatin signaling have been of major interest recently in the NMD field [34]. Myostatin, a member of the transforming growth factor-β superfamily and also known as growth and differentiation factor 8 (GDF-8), is a negative regulator of muscle growth and strength by binding to and activating the receptor complex activin type II (ActRII)/Alk 4/5 (type I receptor) on skeletal muscle [35,36,37]. In IBM, an ActRIIb inhibitory antibody, bimagrumab, increased muscle mass [38] but did not provide clinical benefits in terms of improved mobility [39]. Other myo-/follistatin approaches (for detail see review Nielsen et al., 2021) will need further evaluation in NMD applications [40].

Modification of muscle regeneration is not only of therapeutic interest but should also be considered in preclinical in vitro muscle models to allow the efficient screening. Healthy muscle is an extraordinary tissue that owes the capacity for extensive muscle regeneration to the presence of muscle-resident stem cells (satellite cells) [41]. Muscle regeneration is a tightly regulated stepwise process that can be divided into four consecutive phases: the degeneration phase, the inflammatory phase, the phase of satellite cell expansion and differentiation, and the phase of maturation and remodeling [42,43,44,45]. In all phases, there is a close interplay between immune cells (e.g., T cells, macrophages, and eosinophils), cyto- and chemokine signatures, and satellite or later myogenic cells. Myofibers have been shown to play an important role in the recruitment and activation of immune cells due to the secretion of pro-inflammatory mediators, underlining the relevance of close examination of the interactions between different cell types [46,47].

Satellite cell expansion and differentiation are controlled by the expression of specific transcription factors such as Pax7, MyoD, and myogenin ([48,49] for more details, we refer to the many excellent reviews, e.g., [43]). Of the Pax transcription factors, Pax7 is the predominant regulator during adult muscle regeneration, whereas Pax3 plays a major role in prenatal skeletal muscle formation [50]. A fraction of the activated satellite cells divide asymmetrically with one daughter cell not expressing MyoD, but Pax7 leads to a quiescent population of satellite cells that can become activated when needed [51]. In parallel to the myogenic proliferation-differentiation-maturation-process, immune cells also undergo an adaptation. Similar to the resting satellite cells, dormant leukocytes of the myeloid and lymphoid series are present in the muscle [52,53]. Upon muscle damage, immune cells become activated [54] and within a few hours, neutrophilic granulocytes [55] invade the muscle upon the release of chemokines from the resident macrophages [56]. In addition, resident T cells and the release of chemo- and cytokines such as interferon-γ (IFN-γ) and tumor necrosis factor-α (TNF-α) play an important role in the formation of the proinflammatory milieu [57], which promotes the immigration of further macrophages and T cells. Later in the process, the inflammatory milieu is dominated by interleukin-10 which promotes tissue restoration and especially muscle cell differentiation [58]. Further, satellite and myogenic cells communicate with surrounding cells, in particular, fibroblasts and fibroadipogenic progenitors, to complete maturation and remodeling, thereby enhancing the structural integrity of the muscle fiber including the extracellular matrix and vascularization. Finally, the formation of neuromuscular junctions on regenerated muscle fibers is an important prerequisite for muscle contraction and, therefore, physiological muscle function [59]. Even though the regenerative capacity of skeletal muscle is substantial, the self-repair mechanism can be impaired by various conditions such as large injuries, aging, chronic inflammation, or genetic mutations in muscle proteins [60].

In order to increase the understanding of disease mechanisms, regeneration, drug application pathways, treatment efficacy, and new medication approaches, disease-specific and standardized preclinical models are needed to reach the aim of reducing the disease burden for rare diseases outlined by the European Commission (https://ec.europa.eu (accessed on 27 February 2022)).

## 2. General Considerations for Preclinical Muscle Models

For a dedicated number of hereditary myopathies, animal models are well established and have contributed to innovative drug development such as the *mdx* mouse or additional mouse models with dystrophin mutations more specifically reflecting human Duchenne Muscular Dystrophy [61,62,63,64,65]. However, animal models often show insufficient translatability of obtained results to human physiology and are difficult to interrogate [66]. More than 95% of all animal tested therapeutics fail in clinical studies [67]. For acquired myopathies such as myositis, animal models have increased in numbers over the last decades, but a reproducible model reflecting the phenotypical and histopathological characteristics of IIM is still needed [68].

To reduce the complexity of animal studies and in order to have a scalable and easily controllable model system in muscle research, in vitro skeletal muscle models have been established over decades for muscle and myopathy research. Although rodent cells have been used, it is desirable to apply human muscle cells to better reflect genotype-phenotype correlations, metabolism, and function. What are additional requirements of cell culture systems to be applicable in preclinical drug development? Depending on the stage of development, the requirements of throughput range from high throughput screening (HTS) to a refined analysis of candidates in more complex but physiologically relevant systems. HTS phenotypic screens are being utilized to assess muscle cell structure, size, nucleation, and mitochondrial integrity [69]. While high levels of robustness and reproducibility are desired for the relevance of readouts for clinical response, the prediction has to be carefully evaluated.

The established 2-dimensional (2D) cell culture systems move preclinical research and drug testing forward, but they have three major limitations: (1) cells outside a multicellular network are known to stay in a rather undifferentiated condition, leading to altered metabolic characteristics and restricted analyses of pathomechanisms, (2) the functionality cannot be measured directly, and (3) in vitro disease phenotypes may not reflect the pathophysiology of bonafide patient muscle. Translation of findings in conventional in vitro models towards clinical approaches, especially regarding drug development, therefore remains a challenge and presents a severe obstacle in the bench-to-bedside research pipeline. In summary, in vitro findings are not always applicable to an in vivo environment, since many relevant factors might be excluded from in vitro models. This implies the need for more elaborate human disease model systems.

Keeping this in mind, in this review we will describe the evolutionary state of existing human myogenic co-culture models in two- and three-dimensional structures. We will also highlight the different sources of human muscle cells and their ability to resemble a mature and functional state as well as reflect the inflammatory myopathy entities. Finally, the myo-culture systems will be discussed in the context of their suitability to recapitulate bonafide muscle function and utility in drug testing.

## 3. From 2D Cell Culture to Human Organs-on-a-Chip

While it is possible to isolate human myofibers with more than 36 mm in length from post-mortem muscle tissue [70], primary human skeletal muscle is more regularly accessible via small biopsies. In the latter case, it is exceedingly difficult to obtain intact myofibers for extensive experimentation. To be useful in preclinical studies, a readily available supply of homogenous muscle cells without large batch-to-batch variation is a prerequisite. At the same time, it needs to be considered how well the cellular models reflect the (patho-) physiology of the primary muscle fiber.

### 3.1. Human Muscle Cell Sources

To obtain fit-for-purpose human skeletal muscle cells, different strategies are being applied. The amount of muscle cells required strongly depends on the application and/or experimental approach. Drug screening and disease modeling may be achieved with relatively small numbers if the screening platform can be sufficiently scaled down. However, regenerative strategies or volumetric muscle replacement requires billions of muscle cells.

Classically, human myoblast cultures have been derived by isolation of muscle stem cells from muscle biopsies [71,72]. These muscle stem cells can be readily expanded but may lose stem cell properties in vitro [73]. While muscle stem cell properties may be preserved by hypothermic treatment, culture on pliable substrates, or small molecule addition [73,74,75,76,77], the resulting cell numbers from biopsies are not sufficient to obtain homogenous batches for robust drug screening. In addition, NMD-specific muscle stem cells or muscles from the elderly often exhibit reduced proliferation potential. Primary myoblasts may also vary in their fusion capacity which is a major prerequisite for engineering muscle in vitro [78]. The cellular phenotype and biological function may also change with culture duration and ensuing cellular senescence. Genetic modification to interrogate genotype-phenotype relations is nearly impossible in primary cells. To circumvent some of these obstacles, strategies to make stable, immortalized lines expressing, e.g., SV40 large T antigen or cell cycle activators in combination with telomerase reverse transcriptase have been applied [79,80,81]. While this approach stabilizes proliferation, the ability to fuse into myotubes and karyotypic stability might be affected.

Human induced pluripotent stem cells (iPSC) are another valuable source of skeletal myocytes. A number of efficient directed differentiation protocols have been introduced to derive skeletal myocyte from iPSC [82,83,84,85,86,87]. All of them aim to recapitulate the embryonic development of skeletal muscle in cell culture by temporal modification and simulation of developmental cues. This not only yields a myogenic population of myogenic progenitors, myoblasts, and more mature myotubes but also non-myocytes, e.g., neurons and mesenchymal cells, which may be required for the engineering of physiological muscle in vitro [88]. While this is possible with primary myoblasts using viral vectors, the advantage of pluripotent stem cell-derived myocytes is the ease of genetic modification which ensures robust and reproducible transgene expression. This enables the development of cellular tools that allow the phenotyping of live cells (e.g., by fluorochrome-labeled sarcomeres, Figure 1), light-activation by optogenetic tools, detection of cellular or mitochondrial function by genetic sensors, and genetic correction of disease-associated mutations to obtain isogenic controls.

Another approach is based on seminal findings that transcription factors similar to MYOD1 act as myogenic determination factors that can reprogram a non-myogenic fibroblast into a myoblast [89]. This knowledge is being exploited by programming stem-cell-derived mesodermal cells into myogenic progenitor cells by activation of either PAX7 or MYOD1 transgene [90,91,92]. This results in homogenous myogenic populations which are able to differentiate and fuse into myotubes by by-passing fundamental stages of muscle development [92,93]. A homogenous starting population may be advantageous in disease modeling applications to ensure that phenotypic differences arise from the genotype and not from differences in skeletal myocyte number or quality. However, comparable transgene expression in all tested stem cell lines has to be ensured.

### 3.2. Multi-Cell Type, Multidimensional Approaches to Address NMDs

Skeletal muscle is a complex tissue composed of a multitude of distinct cell types engaging in interlinked processes ensuring muscle function, regeneration, and health. Despite its importance in in vitro research, conventional monolayer monoculture of muscle cells independent of species remains limited regarding the organization, maturation (affecting, e.g., electromechanical coupling), cell type interactions, and functional assessment, e.g., contraction, calcium handling, and muscle force [94,95]. Monolayer co-culture of different cell types, 3D engineered skeletal muscle (ESM) tissue, and neuromuscular organoids can provide research models offering solutions to these constraints and support an improved understanding of neuromuscular disease mechanisms as well as provide highly reproducible options for drug testing (Figure 1).

#### Better Together–2D Monolayer Co-Culture Models

Co-culture systems use two or more cell populations in the same in vitro microenvironment, facilitating interaction among cells in a given cell line population and allowing examination of intercommunication between different types of cells [96]. This provides an experimental model superior in the imitation of the in vivo environment compared to conventional monoculture settings [96,97,98].

Various monolayer co-culture systems of skeletal muscle and manifold other cell lines have been established in the past to study physiological cell interaction as well as disease pathogenesis. This chapter aims to provide an overview of different co-culture approaches and associated disease models (Table 1) as well as shed light on possible clinical relevance and limitations.

Co-cultures of myoblasts with immune cells

Autoimmune phenomena involving immune cells are key players in the development of myositis. Therefore, the interplay between immune cells and myocytes is an important step in disease progression as well as representing a possible target for therapeutic testing. Approaches to co-culture muscle and immune cells, mimicking in vivo conditions of inflammatory neuromuscular disease, date back many years. In 1991, Hohlfield and Engel developed a co-culture system evaluating the effect of T cells isolated from myositis patients’ blood on autologous myotubes resembling the importance of these cells in pathogenesis. They found a lack of autoreactivity in PM and IBM T cell lines, suggesting the presence of additional in vivo factors, such as antigen presentation of myotubes, not sufficiently mirrored by the model [99]. More recently, researchers focused on the influence of T cells on myotoxicity in myositis. A co-culture of CD4^+^ and CD8^+^ CD28(null) T cells, derived from PM patients, with autologous skeletal muscle cells revealed the increased myotoxicity of CD28(null) cells compared to CD28^+^ counterparts with increased susceptibility of myotubes towards cytotoxicity compared to myoblasts. The results led to the conclusion that CD28(null) cells present key effector cells in PM [100]. The basic histopathological changes in PM with evidence of cytotoxic CD8^+^ T cells rearranging and invading the muscle fiber were recreated by Kamiya et al. in 2019 [101] in vitro. They used H2KbOVA-myotubes co-cultured with OT-I CD8^+^ T cells derived from OVA-specific class I restricted T cell receptor transgenic mice and demonstrated the visible invasion of T cells into myotubes. Moreover, invaded myotubes tended to die earlier than non-invaded cells. This co-culture provides a model to further study T cell cytotoxicity in PM but should be supplemented by other factors such as typical myokine signatures.

Another study evaluated the interaction of human dendritic cells (DCs), macrophages, and myoblasts and their effects on myositis [102]. Samples of patients with myositis showed a tendency towards less mature DCs, while myoblasts modulated the degree of maturity of DCs. DCs from the myoblast co-culture proved to have an inhibitory effect on T cell proliferation. Lysates of myoblasts stimulated phagocytose activity of macrophages. Hypothetically, myoblasts could therefore have a modulating function on antigen-presenting cells as a counter-balance to immune mediated muscle damage [102]. Co-culture experiments using immature DCs (iDCs) and LPS-activated DCs (actDCs) combined with human proliferating or differentiating myotubes demonstrated a close interaction of either iDCs or actDCs with muscle cells. Increased muscle proliferation and migration occurred, contrasted by inhibition of muscle differentiation. A stimulating effect of actDCs on HLA-ABC expression and cytokine secretion was apparent as well, promoting an inflammatory environment. The results point towards important interactions between DCs and myoblasts in myositis, interfering with myoblast migration, differentiation and proliferation, and re-feeding ongoing inflammation [17].

Even though these data show an important interplay of a disease-specific immune myo-co-culture, the quantitative and qualitative reproduction of the experiments or their establishment seems to be difficult. To study disease-specific aspects, cells for the co-culture assays are required from one species and in sufficient quantity. In addition, mostly immune cells cannot be directly cultured, but rather need to be activated to establish an interaction with the myotube. Because there are many variables, experiment comparability and high throughput can be difficult to achieve but should precede 3D experiments with reliable results. To circumvent these cellular-based variables, co-stimulatory conditions are chosen by the addition of cytokines and chemokines to produce myositis in in vitro models.

Typical signatures in muscle samples have been detected for different myositis sub-types [11]. Specifically, it was shown for IBM that the muscle fibers attract inflammatory cells expressing cytokines such as IFN-γ, TNF-α, or IL-1β [119], as well as some chemokines [4,120]. In addition, the proinflammatory cytokine IL-1β promoted the IBM-typical β-amyloid accumulation [11] in myotubes [11,121]. How these soluble messengers might act in the 3D model is still poorly understood.

Fat, fibrosis, and muscle

In many muscle diseases, accumulation of the skeletal muscle extracellular matrix occurs alongside muscle wasting and weakness. Disease hallmarks can be fatty infiltration and fibrosis. However, adipogenic cells, as well as fibroblasts, provide unique properties in influencing muscle cell proliferation, differentiation, and regeneration. This interaction between non-contractile cell types and muscle cells in the muscle compartment is studied in co-culture settings.

Fibroadipogenic progenitor cells (FAP) belong to myofiber surrounding cells which contribute to muscle regeneration and are localized in close proximity to blood vessels. FAP possesses a dual role depending on acute or chronic muscle damage [122]. In a mouse model for DMD (*mdx* mouse), they seem to provide the main source of increased fibrogenic or adipose tissue in pathological muscle [123]. Interestingly, a co-culture of fat (3T3-L1) and muscle (L-6) in physical separation but with chemically reciprocal exposure resulted in suppressed lipogenic gene expression and reduced GPDH-activity, suggesting an inhibitory influence of skeletal muscle on adipogenic differentiation [124]. A co-culture of human fat and skeletal muscle cells, obtained from different muscle cell and fat cell donors, showed induction of insulin resistance in human muscle cells through the release of fat cell factors [125,126], underlining the alternating influence of muscle on fat and vice versa.

Different studies have reported an increased rate of muscle differentiation and maturation in a co-culture environment with fibroblasts underlining the relevance of different cell types in muscle cell development and supporting co-culture approaches [127,128]. Myoblasts and fibroblasts isolated from the same mouse muscle, seeded in a fixed arrangement in a cell culture dish providing indirect cell-to-cell contact through an overflow medium, showed enhanced myoblast migration in presence of fibroblasts [129]. Mechanical strain-activated human dermal fibroblasts promoted differentiation of C2C12 myoblasts in a spatially separate co-culture environment through paracrine mechanisms as well [130]. In a transwell-assay to co-culture chicken myoblasts and fibroblasts, fibroblast paracrine factors have been shown to protect differentiating myoblasts from apoptosis [131]. Reconfigurable co-culture devices have been used to examine the influence of paracrine signaling and direct cell to cell contact between C2C12 myoblasts and 3T3 murine fibroblasts singularly or combined, demonstrating a promoting influence of fibroblasts on myoblast alignment only in case of direct contact [132]. This supports the hypothesis that mechanisms surpassing paracrine signaling determine the influence of fibroblasts on muscle cell phenotype and proliferation.

Effects of macrophages and fibroblasts on myoblast proliferation and migration were examined in an in vitro triple co-culture method using mouse C2C12 myoblasts, fibroblasts, and macrophages [133]. In the same publication, a significant increase in myoblast proliferation and migration was detected after co-culture with either unmanipulated macrophages or fibroblasts. Surprisingly, combined with a triple co-culture, it was shown, that the presence of macrophages negated the positive effect of fibroblasts on myoblast migration. Moreover, macrophages lead to an increase in myoblast proliferation, independent of the presence of fibroblasts [133]. The application of an optimized method has revealed quantitative differences in the roles of macrophages versus fibroblasts during alignment and fusion; while successful myoblast alignment is promoted by increasing macrophage numbers, regenerative fusion coincides with a decreasing macrophage population and the initial rise in fibroblast numbers [134]. Therefore, this triple co-culture system highlights the significance of multicellular communication in the regulation of myoblast proliferation and migration and underlines the importance of establishing complex co-culture systems in vitro.

Myo/innervation

Interaction between muscle and nerve presents an important area in understanding neuromuscular disease and in generating more matured and functional cell culture models. The neuromuscular junction is a highly specialized synapse between a motor neuron nerve terminal and the corresponding muscle fiber and is essential for translating neuronal stimuli into muscle contraction. In disorders of neuromuscular transmission, e.g., myasthenia gravis, this connection is disturbed. Various co-culture models employing human or rodent cells address this interplay and show influence on muscle cell growth and maturation through co-culture approaches with neurons [135,136,137,138]. Compared to mono-cell-cultures, increased cell viability of muscle cells in co-culture with neural cells and increased density and length of myotubes were shown [139,140]. One model for the evaluation of molecular events or sprouting of neurites is in the co-culture of C2C12 mouse-myoblasts and PC12 cells (rat phaeochromocytome cells) that differentiate into neurons [112]. PC12 cells possess a synergistic effect on C2C12 differentiation with superior myotube formation which could be instrumental in the development of muscle tissue [141]. These findings support the relevance of neuro-muscular co-culture even beyond the investigation of disorders equivalently involving neural and muscle cells, as physiological muscle can only exist and function in relation to its natural clock generator–the nerve.

A major limitation of monolayer co-cultures is the limited functional read-out possibility regarding muscle contractibility. To overcome this limitation, functional innervation of myofibers using spinal cord explants in co-culture has been established. Human muscle cells can be co-cultured with spinal cord explants of rat embryos with the dorsal spinal ganglia still attached, leading to muscle fiber innervation, contraction, and cross-striation through emerging neurites and neuromuscular junction (NMJ)-formation [142,143,144].

Nevertheless, although monolayer co-culture settings allow for an increased understanding of cellular interactions, no evaluation of the functional implications of disease models, e.g., inflammation on myofibers, is possible. Since impairment of muscle function is a key feature of neuromuscular disease and therefore inhabits a relevant role in disease progression and evaluation of therapeutic success, more advanced in vitro models are needed. Evaluation of muscle strength and contraction force is provided by 3D tissue-engineered muscle.

### 3.3. 3D Muscle Engineering

The generation of skeletal muscle cell culture models with advanced physiological properties has gained much attention recently as it enables not only the testing of molecular markers and cellular morphology but also functional analysis of muscle contraction. Muscle force is one of the most important readouts of muscle function because it integrates cellular, molecular, and metabolic changes. Increasing or restoring contractile function is a primary goal of muscle-targeted therapies. Preclinical developments will especially benefit from it by integrating force measurements into the developmental pipeline.

Several technologies have been introduced to evaluate force production in vitro. Some are based on classical 2D monolayer cultures that use pliable substrates or flexible membranes [108,145]. These “functionalized” monolayers are relatively easy to set up, may be more suitable for higher throughput, and can be scaled down to a single cell/myocyte level. In addition, co-cultures with other cell types can be easily integrated. Even though the cellular morphology is clearly enhanced by directed growth along force axes, functionalized monolayers remain a “flat” culture without 3D configuration.

Tissue engineering of 3D muscle has been introduced more than 30 years ago by Vandenburgh et al. [146]. This approach was recognized early on as a method for durable culturing of muscle cells [146] with reported culture durations of up to 3 months [86]. In addition, 3D muscle engineering may not only improve the cellular morphology of muscle cells but rather allows force measurements similar to primary muscle strips from patients [147]. To obtain functional muscle in vitro, skeletal muscle myoblasts need to be placed into an extracellular matrix in order to mimic a muscular environment. In skeletal muscle, the ECM plays a major role in myofiber stabilization during contraction and regulates muscle regeneration and myogenesis. Specific ECM components are involved in the regulation of satellite cell activity and their modification can influence satellite cell function and consecutive muscle regeneration potential [148,149,150]. The exact composition of the extracellular matrix is of great importance not only for the maturation of muscle cells but also for the functional performance of the engineered muscle [151,152]. In principle, the generation of engineered skeletal muscle can be subdivided into scaffold-based and scaffold-free approaches [153]. Scaffold-based approaches comprise the addition of an artificial synthetic natural extracellular matrix to myoblasts. The most commonly used scaffold substances are hydrogels containing either collagen or fibrin and/or Matrigel^®^. Collagen scaffolds provide high stability and stiffness and were among the first approaches that were used for the generation of 3D muscle [153,154,155,156,157]. A number of labs have turned to fibrin for enhanced tunability and remodeling of the matrix [91,158]. Independent of the primary scaffold, additional extracellular matrix components such as laminin (highly enriched in Matrigel^®^) appear to be critical for skeletal muscle tissue engineering, and the omission of Matrigel^®^ clearly yields suboptimal muscle constructs [153,159,160].

In contrast, non-scaffold-based approaches depend on the extracellular matrix secreted by progenitor cells/myoblasts which then can be enhanced by the addition of growth factors or mechanical stimulation [161]. Here, various techniques exist for the generation of functionally active engineered skeletal muscle including the generation and combination of cellular sheets [162] or through self-assembling myoid/organoid tissues [116,163].

In addition to the cellular microenvironment, external stimuli, namely muscle tension and electrical stimulation, play a major role in muscle differentiation and hypertrophic growth. Vandenburgh et al. already observed in 1979 that muscle tension in vitro increases myofiber diameter [164]. This muscle cell hypertrophy is accompanied by an increase in amino acid uptake and protein synthesis which have recently been reviewed by Ren et al. [165,166]. Electrical stimulation during cell culture can be used to mimic muscle training and induced hypertrophic growth in engineered skeletal muscle together with a strong increase in muscle strength [167,168], indicating the importance of neuromuscular activation during muscle development and emphasizing the need for multicellular in vitro models.

Bioprinting represents an alternative strategy to produce skeletal muscle model systems. Bioprinting is not yet a staple in the field of neuromuscular research but could offer some benefits in the future. If we look into the field of regenerative medicine and research regarding muscle tissue repair, promising results can already be found.

Bioprinting provides the opportunity to build complex constructs through the precise positioning of different cell types, bioactive factors, and biomaterials of different architectural properties mimicking the structure of a specific tissue or organ [169,170,171,172]. For printing live cells, bioinks based on hydrogels have been employed as a carrier substance [173]. In 2016, Choi et al. [174] were able to demonstrate, using decellularized skeletal muscle extracellular matrix (mdECM)-based bioink, that bioprinting allows the development of constructs possessing functional as well as structural features of skeletal muscle. Kang et al., (2016; [175]) presented the integrated tissue organ printer (ITOP) using the principle of printing cell-loaded hydrogels combined with biodegradable polymers in distinct patterns and anchored on a sacrificial hydrogel. The shape was modeled accordingly to tissue-specifics and automatically translated into control of the used printer nozzles applying the cells. Microchannels were implemented to avoid limitations of tissue size through critical diffusion limits. They were able to provide bioprinted models for bone, cartilage, and skeletal muscle. Using this model, Kim et al. (2018; [176]) bioprinted a 3D implantable and biomimetic human skeletal muscle construct using myogenic progenitor cells (MPCs). The construct of an mm³-cm³ scale was used in a rodent model for the reparation of critical tissue defects, showing successful host-nerve innervation and formation of NMJ. Nevertheless, after eight weeks, no full restoration of the defect was reached, probably because of delayed nerve integration in the tissue construct. Effective integration of nerves into donor-muscle tissue remained a challenge. To address this limitation, Kim et al. (2020, [140]) advanced their method further. A bioprinted construct using human muscle progenitor cells (hMPCs) and human neural stem cells (hNSCs), in a ratio of 300:1, provided a neuro-muscular co-culture approach in the bioprinted setting. Results demonstrated increased and fastened maturation and differentiation of myotubes as well as prolonged survival and NMJ formation with Acetylcholine (ACh)-receptor clustering in vitro. The construct was engrafted in a rat model with M. tibialis anterior tissue damage and compared to the previous studies without prior integration of nerve cells; innervation and NMJ-development were improved.

Bioprinting of skeletal muscle constructs provides a promising technique, especially in answering questions regarding regenerative medicine. Further studies on the in-depth analysis of physiological aspects as well as function, especially concerning muscle contractibility, are needed to implement the models into a broader spectrum of research.

#### Co-Culture Approaches in 3D Tissue Engineering

In vitro, 3D muscle imitates in vivo conditions more precisely than in monolayer experiments but is more complex in establishment and may need extensive optimization. To implement the advantages of co-culture settings and 3D engineered muscle tissue, different 3D co-culture approaches have been introduced in the field of neuromuscular diseases. The following chapter concentrates on the principles of co-cultures in a 3D setting.

3D myo/innervation

Regarding the co-culture of skeletal muscle and neural cells especially, recent studies show that neuronal communication has an important influence on muscle formation and will provide an indispensable baseline for future muscle engineering in 3D models [141]. Likewise, skeletal muscle is known to secrete proteins influencing neuronal differentiation [156,177]. Improved muscular contractility and functional outcome have been repeatedly observed in neuromuscular co-cultures [168,178,179]. A co-culture-model of muscle and iPSC-motoneurons in a 3D PDMS (poly(dimethylsiloxane)) scaffold was used to observe the formation of the NMJ in a healthy and neuronal disease model [113]. A functional connection between motoneuron endplates and myofibers was proven. Innervation in 3D neuromuscular co-cultures has been shown to be four times faster and more efficient compared to previously published 2D culture models [180]. Side-to-side comparisons between 2D and 3D co-cultures revealed that the functional integration of adult ACh-receptor epsilon subunit only occurred in the 3D culture setting, underlining the importance of 3D approaches as a more physiological setting beyond functional testing strategies [113]. Additionally, with this study, Afshar Bakooshli et al. (2019; [113]) provided quantitative evidence that a 3D culture environment allows further maturation of multinuclear muscle fibers through experimental inclusion of muscle fiber contractility over longer time periods. This led to muscle fiber hypertrophy, improved calcium homeostasis, and myofiber maturation (e.g., presenting adult MHC-forms and elaborate ACh-receptor cluster). Therefore, 3D neuromuscular cultures are ideal for the examination of synaptogenesis. Moreover, the culture setup was tested as a model for myasthenia gravis (MG) through treatment of the cultures with IgG from MG-patients and a human complement, resulting in a visible clinical phenotype after only two weeks, underlining the possibility of disease-specific usage of this experimental model.

Osaki et al. (2018, [114]) provided an hPSC-derived organ-on-a-chip-model for studying amyotrophic lateral sclerosis (ALS). Firstly, they co-cultured motoneuron-spheroids derived from human embryonic stem cell (hESCs) derived neural stem cells (NSCs) with iPSC-derived myofiber bundles. After separate cultivation and differentiation, co-culture was implemented in a microfluidic device. The growth of neurites, as well as the maturation of myofibers, was observable and neuromuscular junctions formed (NMJ). The functional relevance of NMJs was tested through muscle contraction and Ca²^+^ transients after chemical stimulation of MNs. To model ALS conditions, excitotoxicity was induced through excess glutamate to mimic ALS pathogenesis. In this model, reduced muscle contraction force was detectable, followed by neurite regression and muscle atrophy. Furthermore, iPSC-derived NSCs from a patient with sporadic ALS were co-cultured in a similar approach with iPSC-derived myocytes. ALS-MN with NMJs were detectable after seven days of culture showing the feasibility of this approach. The availability of the model for drug testing purposes was demonstrated by testing rapamycin and bosutinib on ALS and control models, showing prevention of muscle contraction force reduction by day 14 under rapamycin mono- or cotreatment with bosutinib along with neuroprotection. This model represents a reproducible approach with an organ-on-a-chip-technique using patient-derived motoneurons. Perhaps, through the use of patient-derived skeletal muscle cells, the model could be extended to myopathies, providing a platform for the investigation of interactions between muscle and MN activity on the NMJ as well as studies aiming to apply to personalized medicine.

3D myo/inflammation

The approach of the immune-myo-co-cultures can also be transferred into 3D-model systems. In a rat myogenic 3D model, a direct influence of macrophages (bone-marrow-derived rat cells and blood-derived human cells) on increased vessel growth, cell survival, muscle regeneration, and contraction function in muscle has been detected [134]. Cytokine-mediated inflammation in an in vitro tissue-engineered model of human skeletal muscle displayed IFN-γ-dependent myofiber atrophy and contractile loss [103]. Electrical stimulation attenuated this muscle wasting and weakness by the down-regulation of the JAK/STAT1 signaling pathway amplified by IFN-γ. JAK/STAT1 inhibitors fully prevented IFN-γ-induced myopathy as well, confirming the critical role of STAT1 activation in the proinflammatory action of IFN-γ [103].

3D myo/vascularization

Angiogenesis is another relevant aspect of 3D muscle generation in vitro. For example, the construction of thick skeletal muscle still presents a major challenge in tissue engineering. One reason for these limitations is the critical diffusion limit of muscle tissue, currently resulting in a maximum thickness of <1 mm to provide sufficient oxygenation, nutrient supply, and waste disposal. Engineered tissue larger than these limits will result in necrosis of the tissue core. In vivo, the distribution of oxygen and nutrients in the muscle [181] and adaption to the acute needs of the organ are essential for muscle function and health [182,183]. Vascularization and perfusion of the tissue obtain a relevant role in fulfilling those needs [184]. Lacking functional vascularization is therefore a limiting factor of the maximum size of an organoid [185,186]. Furthermore, the interaction between the vascular system and skeletal muscle is of special interest in regard to the pathophysiology of immune cell-mediated neuromuscular disease, therefore, advanced models targeting this interaction are desperately needed.

To approach this challenge, implementation of vessels [187], by including microchannels with HUVEC in collagen in a C2C12 model, was attempted. Self-organization of tissue presents an alternative approach [188]. Skeletal muscle contains non-myogenic supportive tissue such as vascular endothelial cells (ECs) and pericytes (PCs). Their multidimensional function includes organizing blood perfusion simultaneously to control the muscle stem cell compartment homeostasis [189]. The addition of ECs and PCs to engineered muscle could present a physiologically relevant in vitro model to support the survival of larger constructs when fast vascularization is needed to prevent hypoxic cell death, but further research and model development is necessary [187,190,191].

On a different note, angiogenesis is also essential for skeletal muscle regeneration and therefore, is a relevant player in the treatment of neuromuscular disease. Myogenesis and angiogenesis occur simultaneously in muscle regeneration. MPCs are located in proximity to blood vessels and interact with neighboring endothelial cells (ECs) for expansion and differentiation. Latroche et al. (2019, [192]) established a 3D co-culture protocol for the evaluation of MPC activity on angiogenesis and, vice versa, ECs influence on myogenesis, providing an ex vivo assay to improve the understanding of the biological interaction between those cell types in skeletal muscle regeneration. 3D culture studies with myoblasts, fibroblasts, and endothelial cells showed an increase in VEGF secretion compared to a myoblast-endothelial cell co-culture, consistent with the stabilization of the vascular network [158]. Despite those advances, insufficient vascularization remains a major challenge in tissue engineering of muscle. Closely mimicking the vascular system could shed light on the effects of blood flow disruption, disturbance of the blood-muscle barrier (BMB), or invasion of immune cells. An engineered muscle encapsulating a capillary network would allow novel perspectives on the interaction between muscle and blood components in a physiological manner.

The goal to develop a model with an improved vascular compartment was targeted by Bersini et al. (2018; [193]). Myofibers from differentiated myotubes were formed and ECs combined with mesenchymal cells were pipetted into an arch-like structure surrounding the fibers. ECs and bone-marrow-derived mesenchymal stem cells (MCSc) were able to form a microvascular network surrounding the muscle cells. Human muscle fiber-derived fibroblasts migrated towards muscle fibers mimicking the endomysium, while lung-specific fibroblasts did not demonstrate underlining organ-specificity. Fibroblast recruitment was only found in the presence of vasculature. In this model, muscle fibroblasts derived from a DMD-patient were also tested compared to control fibroblasts as well as TGF-β1-treated fibroblasts, a common fibrosis-stimulating in vitro method. In the 3D constructs with DMD-derived fibroblasts, an increased expression of markers of pathological myofibrosis was detectable compared to the control as well as TGF-β1-treated cells, supporting the suitability of this model for studying myofibrosis in muscular dystrophy. Interestingly, in 2D co-cultures, this effect was not visible, underlining the relevance of 3D models for mimicking in vivo conditions superior to 2D monolayer co-cultures.

3D tissue-engineered co-culture models are increasingly used for the examination of muscular dystrophies (Table 1). In a study by Maffioletti et al. (2018, [91]), 3D constructs of skeletal muscle were derived from the hPSCs of healthy donors as well as patients with muscle dystrophy, namely Duchenne, LGMD2D, and LMNA-related dystrophies. The constructs encapsulated distinctive molecular, structural, and functional features of skeletal muscle and proved to be engrafted in immunodeficient mice. In the example of LMNA-related dystrophies, the authors demonstrated distinct disease features mimicked in their constructs, supporting the possibility of engineering disease models derived from donor hPSCs. In a co-culture approach, ECs and PCs were able to be derived from the same hiPSCs used for myogenesis and were combined in a 3D setup. They also included hiPSCs-derived SMI32 + motor neurons but did not deeply characterize the functional impact of those co-cultures. Therefore, this approach provided a stable 3D muscle construct of four isogenic cell types, derived from identical hPSCs, showing muscle-specific as well as disease-related features, demonstrating the possibility of engineering isogenic, multilineage muscle 3D constructs from healthy as well as dystrophic donor hPSCs. This approach opens up the path toward the development of an isogenic human muscle-motoneuron platform which could provide the possibility of modulating neuromuscular diseases in a personalized manner [194,195].

## 4. Neuromuscular Organoids

### What Makes Up an Organoid?

The synonymously used term “mini-organ” already reflects the essential feature of organoids: in a defined three-dimensional matrix, cells differentiate in vitro into a self-organizing, functional multicellular tissue construct that mimics the structure of the respective organ in vivo in a simplified version and is capable of recapitulating its function (in parts) [196]. In this context, the developmental potential of the initial stem cells determines the complexity of the subsequent organoid, whereas the composition of the medium determines which cellular signaling pathways are activated and thus, which organ is mimicked at a microanatomical level.

Embryonic stem cells (ESCs), induced pluripotent stem cells (iPSCs), neonatal, or adult stem cells (ASCs) can be used as starting points for an in vitro generation of organoids.

The use of 3D cultures to model human diseases has become more and more common as a consequence of the development of several protocols that made the long-term expansion of human tissue-generated organoids possible for preserving the original characteristics. One of the important features of this technique is that the structures originated maintain the cell diversity present in the respective organ.

Especially in the field of infection research, human (lung) organoids proved to be of advantage compared to 2D cell culture and animal testing. Previous studies already approved the replication of various respiratory viruses such as the parainfluenza virus 3, measles virus, chikungunya virus, and respiratory syncytial virus for infection in human lung organoids recapitulating native viral infection in humans [197,198]. Further studies involved the analysis of modeling fibrotic lung diseases and malignant and infectious pulmonary diseases in vitro [199,200]. Human tissue-derived organoids open innovative ways for modern medical research to overcome the mentioned limitations and problems when performing studies based on animal experiments and/or 2D immortal cell line culture. Organoids not only provide easy access to investigate, e.g., cell-to-cell or cell-to-pathogen interaction in complex 3D structures, but are suitable for studying molecular processes at the nanometer range as well. Moreover, they allow for high throughput screenings as organoids can be propagated in large amounts even from the same donor.

Principals behind the generation of organoids can be traced back to Steinberg’s differential adhesion hypothesis [201], affirming that different cell types separate based on their adhesion properties. 3D organoids have been developed for several tissues including the intestine, retina, brain, spinal cord, ren, liver, and pancreas [202,203,204,205,206,207,208]. However, the evaluation of diseases affecting more than one tissue remains challenging. Even more, diseases such as muscular dystrophies and motoneuron diseases show secondary effects on NMJs and remain unsusceptible by traditional experimental approaches [209,210]. Organoids modeling skeletal muscle or the neuromuscular apparatus are still rare and need further development, especially in regards to approaches focusing on myo-inflammation or the blood-muscle-barrier. Nevertheless, two recent studies show advances in the generation of human organoids with functional neuromuscular junctions (NMJ) able to stimulate skeletal muscle fibers through activation of neuronal circuits and will be discussed in this review [115,116].

Andersen et al. (2020, [115]), derived organoids resembling the cerebral cortex or the hindbrain/spinal cord from hiPSCs and assembled them with human muscle spheroids for the generation of 3D cortico-motor-assembloids. Regarding the cortico-motor-pathway, species-specific differences in organization and connectivity complicate the translation from animal model to human in vivo settings [211,212,213,214,215], underlining the relevance of complex human in vitro models. Andersen et al. [115] separately generated and then integrated the components of the cortico-motor circuit. They found that human-induced pluripotent stem (hiPS) derived region-specific spheroids form physiologically relevant connections upon assembly. The model remained stable over several weeks and was able to demonstrate cortical controlled muscle contraction, providing significant advances compared to previous 2D co-culture or 3D monoculture settings [180,216,217,218].

A variety of studies instrumented the generation of neuromesodermal progenitor cells (NMPs) from hPSCs for the development of posterior spinal cord neurons and skeletal muscle cells in conventional monolayer cultures [82,219,220,221,222]. Martins et al. (2020, [116]) used hPSC-derived NMPs [220,221,223,224,225] to study the simultaneous development of spinal cord and mesodermal lines in complex 3D organoids. hPSC-derived axial stem cells generated spinal cord neurons and skeletal muscle cells self-assembled to human neuromuscular organoids (NMO). This neuromuscular organoid model system proved highly reproducible between experiments and different PSC lines and showed contractile activity through functional neuromuscular junctions. NMOs from all different hPSC lines examined by Martins et al. started contracting between days 40 and 50, simultaneously with the accumulation of ACh-receptor-clusters in the myofibers. Blockage of ACh-receptors with Curare led to organoid relaxation, suggesting the presence of functional neuromuscular junctions (NMJ). The separation of neural and muscular parts of the organoid was kept intact when on day 50 anatomical and functional NMJs appeared. Terminal Schwann cells, as well as glial cells and myelinisation, were apparent at NMJs. To elucidate whether the neuromuscular disease could be modeled, Martins et al. (2020) treated the neuromuscular organoids on day 50 with autoantibodies retrieved from patients with MG. They showed a significant reduction in the number of Ach-receptor clusters, muscle contraction rate, and amplitude compared to controls. Through the usage of patient-specific iPSCs, this model could prove valuable for research regarding pathogenic mechanisms e.g., in ALS and spinal muscular atrophy (SMA). Evaluation of these diseases still proves to be a challenge resulting from a lack of reliable models portraying critical components of the diseases e.g., Schwann cells. Furthermore, NMOs are electrophysiological active and approachable for functional testing and manipulation. As 3D structures, they are reportedly conservable for more than 1 year and therefore provide access to studying different stages of development of neuromuscular diseases as well as enable the evaluation of the specific roles different cell types play in neuromuscular disease. They can support research concerning drug screening and therapy development [226].

Although the described neuromuscular organoids are not yet specifically targeted toward idiopathic inflammatory myopathies, the possibility of generating complex organoids representing the neuromuscular apparatus presents major progress in the development of in vitro models. Neuromuscular organoids are superior in the evaluation of distinct functions of specific cell types in different stages of neuromuscular junction formation. Maturation of skeletal muscle is more advanced in 3D cultures compared to monolayer cultures as well [113,116]. Further advancement of these models will be relevant to the field of neuromuscular disease research.

## 5. Summary and Outlook

Founding on a large body of work with animal models and animal cells, the development of human-based muscle models has shown great progress in the last couple of years. A variety of in vitro models with varying complexity have been successfully introduced with a strong emphasis on the generation of neuromuscular interfaces. For future directions in the field the following areas of research may be of particular interest:

**Standardization and Validation**: To validate screening platforms and compare readouts, the definition of a set of reference substances would be an important step. The first step for skeletal muscle has been made by setting up a database of Standard Operating Procedures (SOPs) validated by EuroBioBank (http://www.rd-connect.eu) members but does not yet include elaborated protocols for 3D model systems. Concerted actions similar to the Comprehensive in vitro Proarrhythmia Assay (CiPa) consortium to define or validate drug testing activities on neuromuscular model systems or specific clinically relevant readouts may be a viable way forward [227]. In addition, defining positive and negative controls for muscle toxicity testing would be an important asset. A potential list of suggested substances is provided in Table 2**.** Defining a consensus in the field would be an important step forward.

**Vascularization:** The possibility of studying the vascular compartment represents an additional interesting application of muscle organoids. The interaction between endothelial cells and muscle fibers via artificial channels has already been established in 3D cell culture systems [114]. Muscle vascularization plays an important role in the pathogenesis of inflammatory and non-inflammatory muscle diseases. Investigating this interplay at the organoid level may provide valuable pathophysiological insights. Moreover, the general role and functioning of the blood-muscle barrier could be investigated more intensively, and influencing factors may be identified, as it appears to be of great importance for the development of antisense-oligonucleotides or gene therapies for hereditary muscle diseases. The delivery and uptake of gene vectors are essential for gene therapy and could be optimized by the use of complex muscle organoids. Bergmann et al. were able to develop an organoid for the BBB from astrocytes, pericytes, and endothelial cells [228]. This model can be used to study the transport and development of drugs entering the brain. A similar model for vascularized muscle organoids is conceptual and could significantly aid drug development.

**Regeneration:** Several rodents and human models have demonstrated that functional regeneration of skeletal muscle by satellite cells is recapitulated in vitro [86,134,158,229]. It would be important to extend these findings to neuromuscular disease entities to dissect if particular mutations or disease states affect muscle stem cell function in an experimental setting. Another challenge worth addressing will be the development of experimental platforms to allow for drug screening or genetic screens on regenerating muscle in a medium to high throughput format. Ultimately, this may aid the development of pro-regenerative therapies to support the regain of human muscle function.
cells-11-01233-t002_Table 2Table 2List of proposed reference substances to validate neuromuscular in vitro models.TargetSubstanceExpected Effect/ReadoutRef.Neuromuscular junctionCurare (tubocurarine, non-depolarizing muscle relaxants)Ceasing of contraction[168,230]Cholinesterase inhibitors (e.g., neostigmine)Faster recovery of contractile activity in presence of curare[231]L-GlutamateSelective stimulation of neurons[113]Myocyte size/ structureStatinsMyocyte death  Force reduction  Induction of atrophy[135,210,212,213,232]
Corticosteroids  (e.g., dexamethasone)Short term exposure: Increased myogenesis and force production  Long term exposure: atrophy and degeneration[233]
IGF-1Hypertrophy  Increase in tetanic force[69,234]
TNF-αAtrophy[235]
Androgens/selective androgen receptor modulators (SARMs)Hypertrophy  Increase in mTOR/Akt signaling[236]
ClenbuterolConcentration-dependent hypertrophy  Increase in protein content[230,237]
CreatineIncrease in tetanic force production[234]
Myostatin inhibition (e.g., follistatin)Hypertrophy[238]Myocyte functionAcetylcholineMuscle contraction[113,230]
CaffeineRyanodine receptor activation[230]
DantroleneRyanodine receptor inhibition[239]MetabolismInsulinStimulated glucose uptake[240]
MetforminAMPK activation  Increased glucose uptake[240]
ChloroquineInduction of autophagy[230]

## Figures and Tables

**Figure 1 cells-11-01233-f001:**
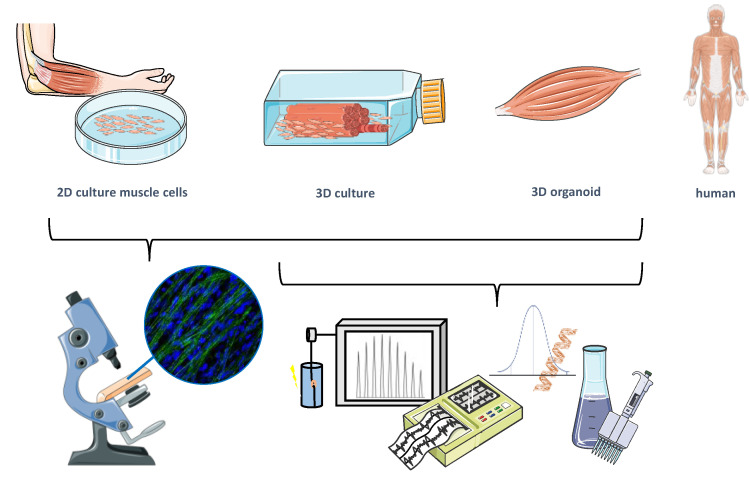
Evolution of cellular tools.

**Table 1 cells-11-01233-t001:** **Established in vitro models of neuromuscular diseases**. Focus on myoinflammatory, muscular dystrophy, and neuromuscular disease entities.

	Cell Types	Cell Origin	Method	Targeted Mechanism	Disease Model	Novelty Findings	Ref.
**Myoin-flammation**	T-cells/skeletal myocytes	Primary cells from PM/IBM patients	Monolayer Co-Culture	Myoinflammation	Myositis (PM, IBM)	Antigen presentation on muscle cells	[99]
	CD4^+^ and CD8^+^ (null) t-cells/autologous skeletal myocytes	Primary cells from PM patients	Monolayer Co-Culture	Myoinflammation	Polymyositis	CD28(null) cells present key effector cells in Polymyositis	[100]
	H2K ^b^OVA- skeletal myocytes/OT-I CD8 ^+^ T cells	OVA-specific class I restricted T cell receptor transgenic mice	Monolayer Co-Culture	Myoinflammation/T-cell cytotoxicity	Polymyositis	Invasion of T-cells into myotubes, death of invaded myotubes prior to non-invaded cells	[101]
	Dendritic cells/macrophages/skeletal myocytes	Primary cells from myositis patients	Monolayer Co-Culture	Myoinflammation	Myositis	Modulating effect of myoblasts on antigen presenting cells	[102]
	Skeletal myocytes	Primary cells from healthy donors	3D myobundle	Myoinflammation/IFN-γ–induced myopathy	Myositis	Direct IFN-γ-induced muscle weakness, counteracted by exercise-mimetic and JAK/STAT inhibitors	[103]
**Inherited myopathies**	Skeletal myocytes	iPSCs derived from DMD patients and control	Monolayer Monoculture	Muscular dystrophy	Duchenne	Morphological and physiological comparable myotubes were able to be differentiated from DMD and control; electric stimulation caused Ca²^+^-overflow only in DMD-myotubes, this was attenuated after dystrophin restoration through exon-skipping	[104]
	Skeletal myocytes	Patient-derived iPSCs and genetic correction	Monolayer Monoculture	Restoration of dystrophin protein	Duchenne	Exon skipping, frameshifting, and exon knock-in; exon knock-in was the most effective approach for dystrophin restoration; iPSC-derived skeletal muscle cells with restored protein expression	[105]
	Skeletal myocytes	iPSCs of patients with Infantile onset Pompe Disease (IOPD)/healthy controls	Monolayer Monoculture	Lysosomal glycogen accumulation through defect of lysosomal acid α-glucosidase (GAA)	Infantile onset Pompe Disease (IOPD)	Lysosomal glycogen accumulation was dose-dependently rescued by rhGAA; mTOR1-activity is impaired in IOPD with disturbance of energy homeostasis and suppressed mitochondrial oxidative function	[106]
	Skeletal myocytes	Human Pompe Disease (PD) iPSCs	Monolayer Monoculture	Lysosomal glycogen accumulation through defect of GAA	Pompe Disease (PD)	Abnormal lysosomal biogenesis is associated with muscular pathology of PD, EB gene transfer is effective as an add-on strategy to GAA gene transfer	[107]
	Skeletal myocytes	iPSCs from DMD patients and corrected isogenic iPSCs	Monolayer Monoculture	Muscular dystrophy	Duchenne	Establishment of a human “DMD-in-a-dish” model using DMD-hiPSC-derived myoblasts; disease-related phenotyping with patient-to-patient variability including aberrant expression of inflammation or immune-response genes and collagens, increased BMP/TGFβ signaling, and reduced fusion competence; genetic correction and pharmacological “dual-SMAD” inhibition rescued the genetically corrected isogenic myoblasts forming multi-nucleated myotubes	[85]
	Skeletal myofibers	Isogenic DMD mutant cell lines	Monolayer Monoculture	Muscular dystrophy	Duchenne	Improved myofiber maturation from human pluripotent cells in vitro; recapitulation of classical DMD phenotypes in isogenic DMD-mutant iPSC lines; rescue of contractile force, fusion, and branching defects by prednisolone	[108]
	Skeletal myocytes	DMD patient-derived iPSC	Monolayer Monoculture	Muscular dystrophy	Duchenne	Generation of contractile human skeletal muscle cells from DMD patient-derived hiPSC based on the inducible expression of MyoD and BAF60C; DMD iPSC-derived myotubes exhibit constitutive activation of TGFβ-SMAD2/3 signaling as well as the deregulated response to pathogenic stimuli, e.g., ECM-derived signals or mechanical cues	[109]
	Skeletal myocytes	DMD patient-derived ESC and iPSC, Primary cells from healthy and DMD patients	Monolayer Monoculture	Muscular dystrophy	Duchenne	Transcriptomic evidence of DMD onset before entry into the skeletal muscle compartment during iPSC differentiation; dysregulation of mitochondrial genes identified as one of the earliest detectable changes; early induction of Sonic hedgehog (SSH) signaling pathway, followed by collagens as well as fibrosis-related genes, suggesting the existence of an intrinsic fibrotic process driven by DMD muscle cells.	[110]
	Skeletal myocytes/ECs/PCs/SMI32+neurons	hPSCs of healthy donors, Duchenne, LGMD2D and LMNA-related dystrophies	3D Co-Culture	Muscular dystrophy	Duchenne, LGMD2D and LMNA-related dystrophies	Stable 3D muscle construct of four isogenic cell types, derived from identical hPSCs; detection of muscle-specific as well as disease-related features,	[91]
	Skeletal myocytes	Primary cells from healthy and DMD patients	Functionalized monolayer	Muscular dystrophy	Duchenne	Studying of muscle formation and function in functionalized monolayer platform using myoblasts from healthy and DMD patients; impaired polarization with respect to the underlying ECM observed in DMD myoblasts; reduced contractile force	[111]
**Neuro-muscular Junction**	C2C12 myoblasts/PC12 cells	-	Monolayer Co-culture	Neuron-muscle interaction	-	PC12 cells possess a synergistic effect on C2C12 differentiation	[112]
	Myofibers/motoneuron	iPSCs	3D PDMS scaffold	Synaptogenesis	Myasthenia gravis (MG)	Functional connection between motoneuron endplates and myofibers was proven; in the 3D setting accelerated innervation, increased myofiber maturation compared to 2D; MG phenotype was inducible	[113]
	Motoneuron-spheroids/myofiber bundles	NSCs/hESCs/iPSC iPSC from a patient with sporadic ALS	Organ-on-a-chip-model	Synaptogenesis, Drug testing	ALS	Formation of functional NMJ; ALS phenotype with reduced muscle contraction force, neurite regression, and muscle atrophy was contrivable; model feasible for drug testing approaches	[114]
	Organoids resembling the cerebral cortex or the hindbrain/spinal cord/human muscle spheroids	iPSC and primary skeletal myoblasts	3D cortico-motor-assembloid	Formation of the cortico-motor circuit	-	Cortical controlled muscle contraction was detectable in hPSC derived specific spheroids through relevant neuromuscular connections upon self-assembly; assembloids were stable over several weeks	[115]
	Spinal cord neu-rons/skeletal myocytes	hPSC	Neuromuscu-lar Organoids (NMO)	Simultaneous development of spinal cord and muscle compartment in complex 3D organoids	MG	First neuro-muscular organoid model-system that proved highly repro-ducible be-tween exper-iments and different PSC-lines and showed con-tractile activi-ty through functional neuromuscu-lar junctions; MG pheno-type was inducible through ex-posure to autoantibod-ies from MG-patients	[116]
	iPSC-derived Motoneurons/skeletal myocytes	iPSC and primary skel-etal my-oblasts	2D cham-bered co-culture sys-tem	Neuron-muscle inter-action	ALS	Integration of motoneurons derived from ALS-patients’ iPSCs and human skele-tal muscle in chambered co-culture system to develop a functional NMJ model providing a platform to study ALS and being adaptable to patient-specific mod-els	[117]
	iPSC-derived Motoneurons/skeletal myocytes	iPSC and primary skel-etal my-oblasts	Chambered co-culture system	Simulation of MG disease mechanisms, drug devel-opment	MG	Functional in vitro MG-model mim-icking reduc-tion in func-tional nA-ChRs at NMJ, decreased NMJ stability, complement activation and blocking of neuromus-cular trans-mission, fea-sible for drug testing	[118]

## Data Availability

Not applicable.

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
