# Peer review of "The Evolution of Complex Muscle Cell In Vitro Models to Study Pathomechanisms and Drug Development of Neuromuscular Disease"

_cells, 2022, doi:10.3390/cells11071233_

Round 1

Reviewer 1 Report

The manuscript of Zschuntzsch et al. is well written and provide a complete overview of the in vitro systems used to study the mechanisms underlying inflammatory myopathies.

However, whereas the description of the in vitro work is clear, the introduction needs to be reconsidered as the critical role of the fiber was underestimated.

major comments:

The five major groups of myopathies discussed in the present review affect differently myofiber type and/or localisation(e.g. perifascicular fibers in dermatomyositis). Moreover, a recent paper (PMID: 32546599) demonstrated the specific signature of the various groups of myopathies. The production of several pro-inflammatory mediators by the muscle fibers was shown as key players in the course of the disease, being at the origin of the pathogenesis. Moreover, regarding this particular role, the fibers contribute at the recruitment and/or activation of the immune cells. Therefore, the introduction needs to replace the fiber at the center of the pathogenesis (e.g. line 81-82, the interplay involved the fiber, not only the immune cells).

Please pay attention at the various typos in the tables.

please insert a citation after the statements lines 61, 317

Minor comments:

'-' inserted in the middle of a word (lines 19, 32,47, 694)

line 238: CD8+(null) should be replaced by CD8+ CD28null

please clarify the sentence line 317. Is it the same work than in ref 102?

Reviewer 2 Report

In the review “The evolution of complex muscle cell in vitro models to study pathomechanisms and drug development of neuromuscular disease” the authors do an extensive analysis of different models available for neuromuscular disorders. The review is well written, covers many topics and give clarity on what’s available at the moment. Moreover, it creates an interesting distinction for the different cell type/support/ and proportions necessary to build a 3D muscular model. The review is relevant, and the literature cited appropriate and the English is very clear. There are just minor comments:

1) When the authors talk about the role of myostatin, there are some references (more accurate about the role of myostatin in muscle growth) to be cited.

2) Reference 17 and 21 do not seem to be relevant to the issue which they refer to

3) When talking about myogenesis especially Pax3 and Pax7 genes, some paper of Prof. Margaret Buckingham group should be included.

4) Line 101-103 is not very clear what the author is trying to state. Please rephrase.

5) Reference 35 does not seem relevant

6) In line 113, the authors write about mouse models with human dystrophin mutations, but from the references it is not understandable which are those models

7) In the paragraph starting line 131, it should be better expressed the concept that it is not easy to test drugs as it not easy to correlate it then with the model.

8) In paragraph 3.1 starting line 153, when talking about human muscle stem cells, the authors should consider adding some papers from Prof. Vincent Mouly group

9) Ref 61 does not seem to be relevant please explain

10) Line 205 to line 208 the concept has been already assessed previously

11) Reference 86: did the author really wanted to put that reference there?

12) When talking about extracellular matrix some paper from Dr. Urciuolo’s group should be considered.

13) When talking about the artificial muscle, it is not well stated anywhere the survival of the colture.

Round 2

Reviewer 1 Report

The manuscript has been sufficiently improved for publication.